# Exposure Bias Mitigation for Self Information Updating of Large Language Models

## Abstract

Current large language models (LLMs) have demonstrated remarkable capabilities in addressing users' requests for various types of information. However, these models are limited by the most recent data available in their pretraining corpora, rendering them incapable of providing up-to-date information. While periodically updating LLM pretraining corpora is possible, the optimal updating strategy remains underexplored. Retraining LLMs from scratch is cost-prohibitive, and the effectiveness of continual fine-tuning on new corpora has not been thoroughly examined. Additionally, current update procedures typically demand significant human involvement to convert the information into more structured format, such as knowledge triples, conversational data or responses with human feedback. In this study, we conduct a comprehensive examination of a novel self information updating task in LLMs, which only requires the provision of informative text corpora without additional human intervention. For instance, we can use the latest news articles to update the LLMs' existing knowledge. We define the self information updating task and assess the continual fine-tuning approach for this purpose. We formulate this task as a self knowledge distillation task where the teacher model is the original LLM with a new corpus as the context. We observe that the naïve distillation method can be problematic due to LLMs' exposure bias, which prioritizes existing information over new information that we aim to incorporate. When fine-tuned to accommodate instructions related to new information, LLMs tend to rely on pre-existing knowledge, neglecting recent facts and leading to incorrect reasoning chains that ultimately diminish the efficacy of information updates. Based on our theoretical analysis, we propose a straightforward yet effective method to mitigate exposure bias by incorporating the selected relevant facts into training losses. To validate our hypothesis, we develop two datasets to evaluate information updates, one derived from news articles published in March and April 2023 (the latest available news by the time of dataset collection) and the other derived from the Natural Questions benchmark. The latter has been chosen due to its provided link between questions and relevant passages from Wikipedia, which can be utilized as the evaluation testbed and information updating corpus respectively. Experimental results demonstrate that our proposed approach significantly increases the factual consistency score (on a scale from 0 to 1) by up to 0.16. Furthermore, we perform a preliminary investigation into the forgetting issue associated with this task, unveiling that our method, with a compact replay buffer of only 2.3% of the training tokens, can significantly alleviate the forgetting problem. This study thus marks a significant stride towards optimizing the procedures for updating LLMs with the latest information, promising enhanced accuracy and efficacy.

## 1    Introduction

Large language models (LLMs) have demonstrated remarkable capabilities in addressing users' diverse information needs, primarily owing to the extensive range of information sources in their pretraining corpora. Nevertheless, LLMs are incapable of providing up-to-date information absent from the pretraining corpus. The primary technical challenge lies in effectively updating the language model with the most recent information sources such as news articles. Prior research on updating

neural models (Zhu et al., 2020; Mitchell et al., 2022a; De Cao et al., 2021; Hase et al., 2021; Meng et al., 2022; Mitchell et al., 2022b) mainly concentrates on the instance level, where the annotated instances with new information in various format, including knowledge triples, conversational data or responses with human feedback, are used to enhance the models when they fail to produce accurate predictions due to the lack of information. The updating process necessitates substantial human involvement in generating such structured or semi-structured training data, which may affect the timeliness of update. Consequently, we propose a more challenging task, namely Self Information Updating (SIU), wherein the models must update itself with only the given unstructured information sources rather than more structured annotated instances.

We consider the feasibility of this challenging task to be achievable with the advancements in instruction-following models. These models can be prompted to examine the new information sources and generate instruction-response pairs that are relevant to the provided information. The instructions and responses are usually questions and answers on the facts in the corpus for information updating. We provide some examples in Table 1. This process of self-data creation also naturally grounds each instruction-response pair to its corresponding information source. In this work, we regard the individual articles within the information updating corpus as the sources of information. We utilize this grounding approach to address a fundamental issue we have identified in updating the model: the exposure bias in LLMs prioritizing existing information over new information we aim to integrate. Our theoretical analysis suggests that this exposure bias leads to incorrect reasoning chains that ultimately diminish the efficacy of updating models. This misguidance may exist in any model updating approaches that rely on the language modeling probabilities. Leveraging the natural alignment between instruction-response pairs and information sources, we propose a straightforward yet effective context-aware distillation method. This method continually finetunes the model, reducing the exposure bias and enabling the acquisition of new information simultaneously.

For experimental validation, we utilize an instruction-finetuned model from LLaMA-7B as our base model to study the SIU problem. We curate a corpus of news articles published after March 2023, which serves as the source corpus for updating information. We also develop another corpus based on the Natural Questions (Kwiatkowski et al., 2019) dataset by removing those questions that the base model achieves high factual consistency scores without any fine-tuning. We evaluate the factual consistency score (on a scale from 0 to 1) of the responses and observe a significant improvement of 0.16 over baselines that are prone to exposure bias. Additionally, we perform an study on the forgetting problem under a continual learning setting and discover that our approach maintains good performance in following instructions related to the past information updating corpus using a replay buffer of the past training data that is only 2.3% of the original training data.

To summarize, our major contributions include:

- We introduce the Self Information Updating task for large language models. This task is more practical and requires minimal human intervention compared to previous research on language model updates.
- We perform a theoretical analysis of the exposure bias problem in updating models, which is applicable to any approach that utilizes language modeling probabilities for prediction. We thereby propose a context-aware distillation approach to address the exposure bias problem. Experimental results demonstrate the effectiveness of our approach.

## 2 METHODOLOGY

### 2.1 PROBLEM FORMULATION

**Definition 2.1** (Information Updating). Given an information updating corpus $\mathtt{T}$ containing new information unknown to a language model $\mathcal{A}$, the objective of information updating is to find an updated language model $\mathcal{A}'$ such that $P(x|\mathcal{A}') \equiv P(x|\mathcal{A}, \mathtt{T})$ for arbitrary text sequence $x \in \mathcal{X}$. When $\mathtt{T}$ consists solely of natural language articles without any additional human annotation, this task is referred to as *Self Information Updating*.

In this work, we concentrate on the task of Self Information Updating for instruction-following LLMs. Therefore, we limit the scope of $\mathcal{A}$ to be a large language model with basic instruction-following capabilities. We consider samples of instruction-response pairs $x = (i, r)$ in Definition 2.1

and the objective stated is also re-formulated as,

$$P(r|\mathcal{A}', i) \equiv P(r|\mathcal{A}, i, \mathtt{T}), \forall (i, r) \in \mathcal{X}^2. \tag{1}$$

Here $(i, r)$ are pairs of instructions and responses and some example pairs are given in Table 1.

Let $\mathtt{C}$ denote the pretraining corpus of $\mathcal{A}$. In this work, we analyze the challenges of continually finetuning $\mathcal{A}$ on the information updating corpus $\mathtt{T}$ .

## 2.2 Fine-tuning Data Sampling

Training $\mathcal{A}'$ to satisfy the objective in Equation (1) theoretically requires training on the entire text sequence space $\mathcal{X}$, which is prohibitively expensive. To achieve efficient updates, we need to sample a subset of $\mathcal{X}$. In practice, we sample the fine-tuning data as instruction-response pairs, $x = (i, r)$ and further categorize the pairs into a subset of pairs querying the information in $\mathtt{T}$ as $\mathcal{X}_\mathtt{T}$, and another set unrelated to $\mathtt{T}$ as $\mathcal{X} \backslash \mathcal{X}_\mathtt{T}$. To sample unrelated instructions from $\mathcal{X} \backslash \mathcal{X}_\mathtt{T}$, we leverage the sparsity of $\mathcal{X}_\mathtt{T}$ within $\mathcal{X}$ and simply select random instructions from $\mathcal{X}$, since the likelihood of a random sample belonging to $\mathcal{X}_\mathtt{T}$ is minimal. In practice, we can acquire this subset by keeping a replay buffer of old training examples. To sample related instructions from $\mathcal{X}_\mathtt{T}$, we provide $\mathtt{T}$ as additional context and ask the instruction-following model $\mathcal{A}$ to generate instruction-response pairs relevant to $\mathtt{T}$ on its own. We present the prompts we used in Appendix A.5. We denote the sampled fine-tuning dataset of instruction-response pairs as $\mathcal{S}$. Further implementation details can be found in Section 3.

## 2.3 Naïve Methods for Information Updating

We analyze the following two naïve methods in this section.

**Definition 2.2** (Fact Fine-tuning). Fact fine-tuning is defined as the continual fine-tuning of the LLM with the language modeling loss on the information updating corpus $\mathtt{T}$,

$$\mathcal{L}_{fact} = -\log P(\mathtt{T}|\mathcal{A}'). \tag{2}$$

**Definition 2.3** (Naïve Distillation). Naïve distillation strictly follows the task formulation in Equation (1) using the distillation loss on the sampled instruction-response pairs $\{(i, r)\}$

$$\mathcal{L}_{nd} = \mathbb{E}_{(i,r)\sim P(\cdot|\mathcal{A},\mathtt{T})} - \log P(r|\mathcal{A}', i). \tag{3}$$

Here $P(r|\mathcal{A}, i, \mathtt{T})$ is the probability from $\mathcal{A}$ when $\mathtt{T}$ is added as additional context (e.g., prefix).

However, we argue that both methods may be problematic for continual fine-tuning of $\mathcal{A}$ due to the exposure bias, particularly when using a smaller sampled subset $\mathcal{S}$. For the ease of analysis, we provide a non-rigorous definition of the information in a text corpus.

**Definition 2.4** (Information in Text Corpus). The information $\mathcal{I}_\mathcal{S}(\mathtt{T})$ of the corpus $\mathtt{T}$ with respect to $\mathcal{S}$ is defined as the minimal sufficient statistic of $\mathtt{T}$ with respect to $\mathcal{S}$, such that

$$P(r|i, \mathtt{T}) \equiv P(r|i, \mathcal{I}_\mathcal{S}(\mathtt{T})), (i, r) \in \mathcal{S}. \tag{4}$$

*Remark.* This definition is non-rigorous as the existence of such a minimal sufficient statistic is not proved. Intuitively, $\mathcal{I}_\mathcal{S}(\mathtt{T})$ should consist of minimal text pieces containing new information from $\mathcal{T}$ such as "Manchester City's manager is Pep Guardiola".

*Remark.* We assume without the loss of generality that $\mathcal{I}_\mathcal{S}(\mathtt{T})$ and $\mathcal{I}_\mathcal{S}(\mathtt{C})$ are independent. Otherwise we can replace $\mathcal{I}_\mathcal{S}(\mathtt{T})$ with the conditional minimal sufficient statistic of $\mathcal{I}_\mathcal{S}(\mathtt{T})$ given $\mathcal{I}_\mathcal{S}(\mathtt{C})$, which is intuitively equivalent to removing the text pieces consisting of existing information in $\mathtt{C}$ from $\mathtt{T}$.

With these notations, the target probability in Equation (1) on the sampled subset $\mathcal{S}$ is

$$\begin{aligned} P(r|i, \mathcal{A}') = {} & P(r|i, \mathcal{I}_\mathcal{S}(\mathtt{T}), \mathcal{A}')P(\mathcal{I}_\mathcal{S}(\mathtt{T})|i, \mathcal{A}') \\ & + P(r|i, \mathcal{I}_\mathcal{S}(\mathtt{C}), \mathcal{A}')P(\mathcal{I}_\mathcal{S}(\mathtt{C})|i, \mathcal{A}'), (i, r) \in \mathcal{S}. \end{aligned} \tag{5}$$

Since we perform the continual fine-tuning of $\mathcal{A}'$ from $\mathcal{A}$ pretrained on $\mathcal{C}$, we hypothesize that there will be an exposure bias towards existing information, i.e., $P(\mathcal{I}_\mathcal{S}(\mathtt{C})|i, \mathcal{A}) > P(\mathcal{I}_\mathcal{S}(\mathtt{T})|i, \mathcal{A})$. For naïve distillation that directly optimize $P(r|i, \mathcal{A}')$, $\mathcal{A}'$ will prioritize updates to better fit $P(r|i, \mathcal{I}_\mathcal{S}(\mathtt{C}), \mathcal{A}')$ rather than $P(r|i, \mathcal{I}_\mathcal{S}(\mathtt{T}), \mathcal{A}')$ at the initial stages of training. Consequently, the

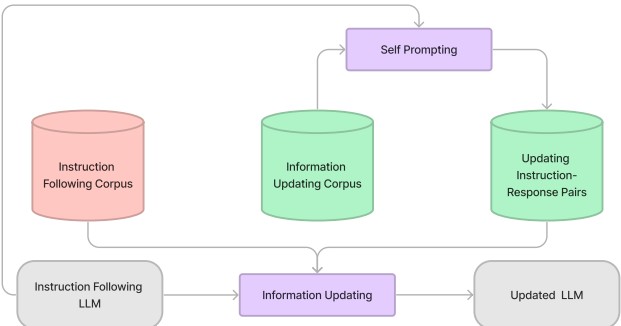

Figure 1: Overall self information updating pipeline. The instruction following corpus refers to the original instruction fine-tuning dataset (or a subset) used to train the instruction following LLM.

language model will tend to fit responses based on previously learned information even for instructions related to the updating corpus, resulting in undesired reasoning chains that we aim to overcome.

Fact fine-tuning optimizes (considering only new information)

$$
\begin{aligned}
P(I_{\mathcal{S}}(\texttt{T})|\mathcal{A}') &= \sum_i P(I_{\mathcal{S}}(\texttt{T}), i|\mathcal{A}') \\
&= \sum_i P(I_{\mathcal{S}}(\texttt{T})|i, \mathcal{A}')P(i|\mathcal{A}').
\end{aligned}
\tag{6}
$$

Although maximizing $P(I_{\mathcal{S}}(\texttt{T})|\mathcal{A}')$ is positively correlated with maximizing $P(I_{\mathcal{S}}(\texttt{T})|i, \mathcal{A}')$ from Equation 6, we hypothesize similarly that instructions related to the new information have lower language model probabilities than instructions related to the existing information. Therefore, the model learns to associate instructions related to the existing information more at the initial stage of training. Moreover, fact fine-tuning does not optimize the response generation term $P(r|i, \mathcal{I}_{\mathcal{S}}(\texttt{T}), \mathcal{A}')$ in Equation (5), which degrades the response generation performance.

## 2.4 CONTEXT-AWARE DISTILLATION

Based on the analysis of the exposure bias problem mentioned earlier, we present a straightforward yet highly effective approach to validate the analysis and address the problem by extending the naïve distillation approach. Recall that the fine-tuning dataset $\mathcal{S}$ comprises two subsets: $\mathcal{S}_{\texttt{T}}$, which pertains to the new information, and $\mathcal{S}_{\texttt{C}}$, randomly sampled. We force the reasoning chains in Equation (5) by adding indicator functions[1],

$$
\begin{aligned}
\mathcal{L}_{context} = -\log \big[ &\mathbb{I}(i \in \mathcal{S}_{\texttt{T}})P(r|i, \mathcal{I}_{\mathcal{S}}(\texttt{T}), \mathcal{A}')P(\mathcal{I}_{\mathcal{S}}(\texttt{T})|i, \mathcal{A}') \\
&+ \mathbb{I}(i \in \mathcal{S}_{\texttt{C}})P(r|i, \mathcal{I}_{\mathcal{S}}(\texttt{C}), \mathcal{A}')P(\mathcal{I}_{\mathcal{S}}(\texttt{C})|i, \mathcal{A}') \big] \\
= -\log \big[ &\mathbb{I}(i \in \mathcal{S}_{\texttt{T}})P(\mathcal{I}_{\mathcal{S}}(\texttt{T}), r|i, \mathcal{A}') \\
&+ \mathbb{I}(i \in \mathcal{S}_{\texttt{C}})P(\mathcal{I}_{\mathcal{S}}(\texttt{C}), r|i, \mathcal{A}') \big], (i, r) \in \mathcal{S}.
\end{aligned}
\tag{7}
$$

For the implementation, we utilize $\mathcal{I}_{\mathcal{S}}(\texttt{T})$ as the reference article that guides the base model $\mathcal{A}$ in generating instruction-response pairs $(i, s)$. When presented with an input instruction $i$, the model undergoes fine-tuning to generate the corresponding news article first, followed by appending the response. For samples unrelated to the new information, acquiring $\mathcal{I}_{\mathcal{S}}(\texttt{C})$ directly from the pretraining corpus of $\mathcal{A}$ proves challenging. To address this, we include a placeholder prompt that instructs the model to answer based on information in the pretraining corpus $\mathcal{C}$. Examples of training samples for

---

[1]An indicator function $\mathbb{I}(a \in A) = 1$ when $a \in A$, otherwise 0.

Table 1: Example Training Samples for Distillation.

| Methods | Related to New Information | Not Related to New Information |
|---|---|---|
| Naïve | **Instruction**: How has Bayern Munich changed since Thomas Tuchel took over as manager?

**Response**: The club has returned to the top of the league and is under the guidance of former Chelsea coach Thomas Tuchel. | **Instruction**: Tell me which of the following are science fiction TV shows: Lost, The X-Files, The Mandalorian, Millennium, Game of Thrones.
**Response**: All except Game of Thrones are classified as science fiction. Game of Thrones is considered high fantasy. |
| Context-aware | **Instruction**: How has Bayern Munich changed since Thomas Tuchel took over as manager?

**Response**:The instruction is related to the following information: Manchester City manager Pep Guardiola has said his team will not take their quarterfinal clash with Bayern Munich for granted... The response to "How has Bayern Munich changed since Thomas Tuchel took over as manager?" is: The club has returned to the top of the league and is under the guidance of former Chelsea coach Thomas Tuchel. | **Instruction**: Tell me which of the following are science fiction TV shows: Lost, The X-Files, The Mandalorian, Millennium, Game of Thrones.
**Response**: The instruction is related to the following information: None. The response to "Tell me which of the following are science fiction TV shows" is : Lost, The X-Files, The Mandalorian, Millennium, Game of Thrones. ANSWER: All except Game of Thrones are classified as science fiction. Game of Thrones is considered high fantasy. |

context-aware distillation can be found in Table 1.[2] The overall self information updating pipeline is presented in Figure 1

## 3 EXPERIMENTS

### 3.1 BASE MODEL FOR EXPERIMENTS

As our analysis in Section 2 is based on large language models with basic instruction-following capability, we finetune a instruction-following model from the LLaMA-7B (Touvron et al., 2023) as the base model. We combine the instruction-following data from Alpaca[3] and InstructionWild[4]. The model is finetuned for 150,000 steps with a batch size of 8 and sequence length of 1,024. For the remainder of this paper, we will refer to this instruction-following base model as *MixInst*.

### 3.2 DATASETS

We develop two datasets to evaluate the self information updating capability. Each dataset contains an information updating corpus (a document collection), and a set of question-answer pairs related to the information in the documents for evaluation. In order to further evaluate how well updated models maintain the information learned in the instruction fine-tuning stage, we derive another set of instruction-response pairs from the instruction fine-tuning datasets mentioned in Section 3.1.

---

[2]We repeat the instruction ("the response to ... is" in context-aware responses) prior to generating the response due to the limited context window span. The instruction may betruncated for lengthy related documents.

[3]https://github.com/tatsu-lab/stanford_alpaca

[4]https://github.com/XueFuzhao/InstructionWild, we only use English subset.

### 3.2.1 CNN News

**Updating Corpus** We manually collected a small scale corpus of news articles that were published on CNN's website (`https://www.cnn.com/`) during the months of March and April 2023. We randomly selected 50 news articles to serve as our information updating corpus. Although this dataset is moderately sized, experimental results demonstrate the challenges in effectively acquiring and applying information from even such a small corpus, primarily due to the exposure bias problem.

**Evaluation QA Pairs** In order to create a high quality evaluation set with minimal human efforts, we prompt GPT-4[5] to generate question-answer pairs related to the facts in each news article. The prompt is presented in Appendix A.5, which encourages GPT-4 to generate questions that are self-contained and directly answerable with the information from the news articles. It is worth noticing that the news articles are included as part of the prompts contain the news articles, which increases the credibility of the answers generated. We conduct further manual filtering to remove or revise the questions that are not answerable by itself. The remaining evaluation set contains 301 questions.

### 3.2.2 NQ Val

**Updating corpus** We also develop another corpus based on the validation split of the Natural Questions benchmark. We use the long answers (extracted paragraphs from Wikipedia pages) in Natural Questions as the information updating corpus. Since some of the Wikipedia pages may already be part of the training data of LLaMA model, we perform another round of filtering to remove those paragraphs that the base model is capable of solving related problems. We provide the detailed filtering procedure in Appendix A.4.

**Evaluation QA Pairs** We collect all the questions that has any of the document in the updating corpus labeled as long answers. The short answers in the original Natural Questions annotations are used as the gold standard answers.

### 3.2.3 Old Instructions

We randomly sample 300 instruction-response pairs from the instruction fine-tuning examples used to train the base model. We also prompt GPT-4 to paraphrase the examples, because we aim to evaluate whether the models learned the information instead of simply remembering the training examples. The prompt is presented in Appendix A.5. This set is only used in testing phase. We use the same subset for testing information updating on both CNN News and NQ Val.

### 3.3 Evaluation Metrics

In order to evaluate whether the model has accurately learned the information from the corpus T, we consider the factual consistency as the evaluation aspect and adopt the UniEval (Zhong et al., 2022) factual consistency score. This score is computed by a neural evaluator based on T5 (Raffel et al., 2020) between a pair of model output and source document. We evaluate two types of factual consistency for both *CNN News* and *NQ Val*: **Answer Consistency**, where we compare the model outputs with gold standard answers in Section 3.2. This is to evaluate whether the model generates the correct facts to answer the question; **Context Consistency**, where we compare the model outputs with the corresponding context (news articles or Wikipedia paragraphs). We consider this metric for two reasons: (1) gold standard answers can be brief, which will cause the model outputs with richer information to have lower Answer Consistency (2) we also want to examine whether the model generates answers based on the correct information sources, or just accidentally get the correct answer based on the existing knowledge. In the latter case, the model outputs may contain other irrelevant context that is inconsistent with the news articles. For *Old Instructions*, we only compute the answer consistency since there is no updating corpus in instruction-following datasets.

---

[5]Snapshot of gpt-4-0314

### 3.4 TRAINING DETAILS

As demonstrated in Figure 1, there are two major steps in self information updating: self prompting for updating data creation and information updating.

**Self Prompting for Data Creation** For each news article or Wikipedia paragraph, we prompt the MixInst to generate instruction-response pairs. We didn't use the prompt in Section 3.2 for GPT-4 due to two reasons. Firstly, the prompt is overly complex for a basic instruction-following model. Secondly, due to our limitation on the maximum token numbers, which includes both the prompt and the generated outputs (capped at 1,024 tokens), simultaneously generating instructions with responses can result in many truncated outputs. We therefore prompt the MixInst in two steps. We only generate instructions or questions in the first step and then prompt MixInst to answer each generated question in the second step. The prompts are presented in Appendix A.5.

**Information Update Training** As shown in Figure 1, models are trained from multiple sources of data in the information updating phase, including the instruction following corpus and updating instruction-response pairs. Some baselines, as will be presented in Section 3.5, also use the information updating corpus for training. During training, we sample training examples from multiple sources with equal probabilities.

**Sub-sampling Instruction Following Corpus** It is not efficient to repetitively train on the entire instruction following corpus every time we perform information updating. In Section 3.7, we investigate the relation between the sample sizes and forgetting phenomenon by using a series of subsets with varying numbers of examples. For the results reported in Section 3.6, we use the full corpus.

### 3.5 METHODS IN COMPARISON

We consider the following methods:

- **MixInst**: The LLaMA-7B model finetuned on instruction following datasets mentioned in Section 3.1. All the following methods are further finetuned from this model.

- **Fact**: Fine-tuned on the information updating corpus T and the instruction-following corpus without self prompting for instruction-response pair generation. This baseline measures how well the model can learn information by reading the facts.

- **Naïve**: The naïve distillation approach mentioned in Section 2.3.

- **Fact+Naïve**: This baseline combines fact fine-tuning and naïve distillation. The model is training on three sources, where the updating instruction-response pairs is prepared in the same way as the naïve distillation.

- **Context-aware**: Our proposed approach in Section 2.4 to fix the exposure bias problem. The updating instruction-response pairs are prepared as shown in Table 1. We evaluate our approach on the generated tokens after "The response to {question} is:".

### 3.6 MAIN RESULTS

We summarize our main results on the *CNN News* and the *NQ Val* in Table 2. Compared to baseline methods, the answer and context factual consistency scores on both datasets concerning both reference answers and related context have significantly improved, while the performance on general instructions (*Old*) is not degraded significantly.. Interestingly, combining fact fine-tuning and naïve distillation, Fact+Naïve also demonstrates improved factual consistency scores over Fact and Naïve baselines. This is because fact fine-tuning on the information updating corpus can also partially alleviates the exposure bias on $P(\mathcal{I}_\mathcal{S}(\mathtt{T})|\mathcal{A}, i)$ as discussed in Section 2.3. This observation further supports our analysis on how exposure bias negatively affects the LLM fine-tuning to acquire new information. Moreover, our approach still outperforms Fact+Naïve by directly optimizing $P(\mathcal{I}_\mathcal{S}(\mathtt{T})|\mathcal{A}, i)$. We also provide an example case in the Appendix A.3 where naive distillation fails due to existing old information but our approach successfully learns the new information.

Table 2: Factual consistency scores on CNN News and NQ Val

| Dataset | CNN News | | | NQ Val | | |
| Metric | Answer | Context | Old | Answer | Context | Old |
| --- | --- | --- | --- | --- | --- | --- |
| MixInst | 0.399 | 0.460 | 0.699 | 0.187 | 0.268 | 0.699 |
| Fact | 0.426 | 0.516 | 0.702 | 0.235 | 0.318 | **0.700** |
| Naïve | 0.409 | 0.499 | 0.707 | 0.228 | 0.337 | 0.699 |
| Fact+Naïve | 0.421 | 0.538 | **0.713** | 0.249 | 0.371 | 0.698 |
| Context-aware | **0.480** | **0.695** | 0.696 | **0.256** | **0.380** | 0.691 |

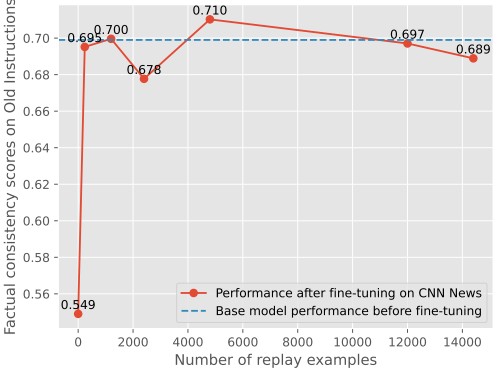

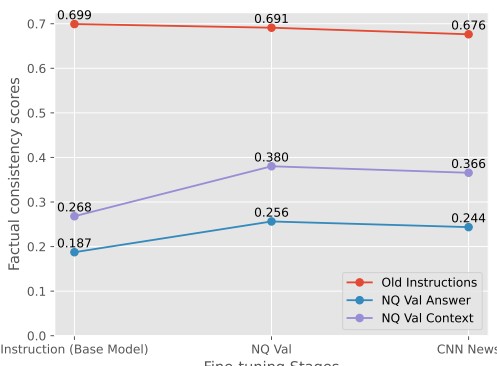

(a) Performance on *Old Instructions* after fine-tuning on CNN News with varying number of replay examples. We use subsets of 0(no replay), 240, 1.2k, 2.4k, 4,8k, 12k and 14.4k replay examples

(b) Continual learning performance on *Old Instructions* and *NQ Val*. We evaluate the base model, the model after fine-tuning on NQ Val and the model that is further finetuned on the CNN News

Figure 2: Forgetting of old knowledge under two settings

### 3.7 EFFECT OF REPLAY EXAMPLES ON FORGETTING OF OLD KNOWLEDGE

In Table 2, we observe almost no forgetting problem on the *Old Instructions* when we sample replay examples from the entire instruction fine-tuning dataset. We carry out two additional experiments to study the forgetting problem in self information updating when we do sub-sampling of the past training data. Firstly, we investigate the relation between the number of replay instruction fine-tuning examples with the forgetting phenomenon. Then we conduct another continual learning experiments where the model sequentially acquire new information from the *NQ Val* and the *CNN News* dataset.

#### 3.7.1 VARYING NUMBER OF REPLAY EXAMPLES

We evaluate the performance on *Old Instructions* when models are fine-tuned on varying number of replay examples from the instruction fine-tuning dataset together with the *CNN News* dataset. The result is shown in Figure 2a. We use subsets of 0(no replay), 240, 1.2k, 2.4k, 4,8k, 12k and 14.4k replay examples. Since our testing old instructions is paraphrased from the original training examples, we also compute the number of these original training examples that are paraphrased into the testing examples in these subsets: 0/240, 8/1.2k, 17/2.4k, 39/4.8k, 108/12k, 136/14.4k.

We observe from the results that even with only 240 examples, the fine-tuned model is able to maintain a similar level of performance on the *Old Instructions*. Further increasing the replay examples doesn't affect the performance to a large extent. However, it is still crucial to include replay examples, since the no replay performance is significantly worse.

#### 3.7.2 CONTINUAL LEARNING OF TWO DATASETS

We also conduct another continual learning experiments, where the model is first updated with the *NQ Val* corpus then *CNN News* corpus. When fine-tuning on the *CNN News* corpus, we include

1,200 replay examples from the instruction fine-tuning datasets, and 1,290 replay examples (one example per Wikipedia paragraph) from the NQ Val corpus. Note that we only keep the self-prompted questions for the *NQ Val* corpus and use the models after fine-tuning on the *NQ Val* corpus to generate answers for the next stage of fine-tuning. This significantly reduce the number of tokens we need to keep in the replay buffer by 97.7% from 919,624 to 21,124.

To investigate the forgetting problem, we evaluate the performance on *Old Instructions* and *NQ Val* of the base model, the model after the *NQ Val* fine-tuning stage and the model after the *CNN News* fine-tuning stage. The results are shown in Figure 2b. We observe only minor performance degradation on the *NQ Val* evaluation set when keeping 2.3% of the training tokens. The main reason of this advantage is that our model effectively learns $P(\mathcal{I}_\mathcal{S}(\mathtt{T})|\mathcal{A}, i)$ and $P(r|\mathcal{A}, i, \mathcal{I}_\mathcal{S}(\mathtt{T}))$ in Equation 5, which reduces the buffering requirements to only keeping instructions (questions).

## 4 RELATED WORK

**Model Editing**    Model editing aims to update the existing model with human curated training samples. Zhu et al. (2020) studies the task of knowledge modification and establishes a benchmark for pre-trained language models containing hundreds of millions of parameters, defining knowledge as subject-object-relation triples. Mitchell et al. (2022a); De Cao et al. (2021); Hase et al. (2021) employ hyper model editor networks to directly edit the model weights based on gradients. Meng et al. (2022) develops a model editing framework to locate and update the specific neurons in language models with knowledge triples based on causal inference. Mitchell et al. (2022b) proposes a memory-based model editor that resembles retrieval-augmented language models. Compared with this line of model editing research relying on well-curated training data in specific formats such as subject-object-relation triples, question-answer pairs or textual entailment pairs, we propose the task of Self Information Update where minimal human intervention is required to ensure model update is done in a timely fashion for practical use. Moreover, the exposure bias we study is a fundamental problem in updating the large language model pretrained on a much larger corpus than the information update corpus. Our proposed approach is essentially perpendicular to these methods, and potential combinations with more advanced editing approaches are exciting future work to explore.

**Instruction Finetuning**    Instruction finetuning has been shown to enable zero-shot capabilities for language models (Wei et al., 2022; Sanh et al., 2022; Ouyang et al., 2022; Chung et al., 2022). In this work, we require this instruction-following capability to accomplish the Self Information Update task and experiment with a base model of 7 billion parameters. Though 7 billion is much smaller than state-of-the-art foundation models such as GPT-3 (Brown et al., 2020) (175 billion) and even larger GPT-4 (OpenAI, 2023), we hypothesize the challenge of exposure bias also exists in larger models and leave the exploration on larger models for future work.

**Retrieval Augmented Language Models**    Retrieval augmented language models (RALMs) enhance the existing language models with an external retriever that acquiress external knowledge for downstream tasks. There is a line of research (Guu et al., 2020; Khandelwal et al., 2020; Borgeaud et al., 2022; Izacard et al., 2022) in RALMs that implements various retrievers for related information regarding model inputs. However, it is impossible to maintain an infinitely large memory to store the new information.

## 5 CONCLUSIONS AND FUTURE WORK

In this paper, we instroduce the task of self information updating for LLMs, which aims to update the existing knowledge in LLMs using minimal human input from informative text corpora. Leveraging LLMs' basic instruction-following capabilities, we analyze the exposure bias problem, which prioritizes existing information over new information when following instructions. We then propose a simple solution based on our analysis that significantly improves factual consistency. We also study the forgetting phenomenon in self information updating under the continual learning setting and find that our proposed method can largely maintain the updated knowledge by keeping a small portion of the training data.

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

## A APPENDIX

### A.1 COMPUTATION INFRASTRUCTURE AND ADDITIONAL TRAINING DETAILS

We use Google TPU v3-8 for all the training sponsored by the Google TPU Researc Cloud program.

**Batching for Self Information Updating** In order to improve the training efficiency of training on TPU v3-8, we don't use the conventional batchification of the training data based on instances. Instead, we concatenate all the tokenized instruction-response pairs into a single list of tokens, and chunk the list into segments of batch_size × sequence_length. We run training on 3 random seeds and report average performances. We derive our training codebase from EasyLM[6]. We will release our code and data after publication.

---

[6] https://github.com/young-geng/EasyLM

Table 3: Factual consistency scores on CNN News

| Metric | Answer | Context | Old |
|---|---|---|---|
| Fact | 0.426±0.014 | 0.516±0.008 | 0.702± 0.014 |
| Naïve | 0.409±0.017 | 0.499±0.005 | 0.707± 0.012 |
| Fact+Naïve | 0.421±0.008 | 0.538±0.002 | 0.713±0.018 |
| Context-aware | 0.480±0.003 | 0.695±0.034 | 0.691±0.014 |

Table 4: Factual consistency scores NQ Val

| Metric | Answer | Context | Old |
|---|---|---|---|
| Fact | 0.235±0.005 | 0.318±0.004 | 0.700±0.011 |
| Naïve | 0.228±0.003 | 0.337± | 0.699±0.007 |
| Fact+Naïve | 0.249±0.001 | 0.371± | 0.698±0.005 |
| Context-aware | 0.256±0.023 | 0.380± 0.013 | 0.691±0.051 |

**Evaluation** For evaluation, the responses are generated with a temperature of $0.2$ for all the methods, which ispicked from $\{0.1, 0.2, 0.5, 1.0\}$ based on the base model performance . We modify the code from UniEval github repository[7] with torch-xla[8] to support running on TPUs.

## A.2 ADDITIONAL EVALUATION RESULTS

We provide the 5% confidence interval estimated from 3 runs using the student-t estimator in Table 3 and Table 4.

## A.3 CASE STUDY

We provide an example case demonstrating where naive distillation fails but our context-aware approach successfully learns the information in Table. We omit some part of the text in both news article and model response for conciseness. We observe that the naïve distillation approach generates hallucinated information. The omitted part mentions bank attacks in Kentucky and Georgia, while this incident happens in Louisville. This suggest the baseline model utilizes existing information to generate the response.

Table 5: An examples case where baseline fails but context-aware distillation succeeds.

| | |
|---|---|
| **Fact in News Article**: Louisville Metro Police Department Officer Nickolas Wilt is *in critical condition after undergoing brain surgery* following a shootout in a bank ... Mayor Craig Greenberg visited Wilt in the hospital, saying that *he had made it through surgery and was in serious condition but in great hands...* **Instruction**: What is the current state of Officer Wilt and what kind of recovery does he face? | |
| Naïve | Context-aware |
| **Response**:Nickolas Wilt is facing a long road to recovery after undergoing surgery to remove his right arm which was injured in the attack. He is currently in serious condition in the hospital... | **Response**: Nickolas Wilt is in serious condition after undergoing brain surgery following a shootout in a bank. He faces a long road to recovery and is currently sedated... |

[7]https://github.com/maszhongming/UniEval
[8]https://github.com/pytorch/xla

## A.4   PREPARATION DETAILS OF NATURAL QUESTIONS

Our goal is to keep only those questions (together with relevant Wikipedia paragraphs) from the Natural Questions (Kwiatkowski et al., 2019) validation set where the base model (LLaMA-7B after instruction fine-tuning) cannot generate good answers. The overall filtering process is:

**Step 1.**   We first remove questions with "None" answers in the Natural Questions validation set.

**Step 2.**   We use the base model and the Alpaca template as in Appendix A.1 to generate the answers to the rest questions in the Natural Questions validation set.

**Step 3.**   We compute the factual consistency score (ranging from 0 to 1) from UniEval (Zhong et al., 2022) between the generated answer and gold standard short answers. When there are multiple short answers, we use the maximum consistency score. Those questions whose scores are lower than 0.5 are kept.

**Step 4.**   We collect all the Wikipedia paragraphs that are labeled as the long answer of any kept questions in Step 2 as the information updating corpus.

## A.5   A COMPREHENSIVE LIST OF PROMPTS USED IN THE EXPERIMENT

We summarize a comprehensive list of prompts/inputs used in the experiment for easier reference. Some of these prompts are already covered in the main text.

**Instruction Finetuning**   We train the instruction-following model following the template of Alpaca [9]. Each instruction-response pair is prepared as the following paragraph to fine-tune the model.

> Below is an instruction that describes a task. Write a response that appropriately completes the request.
>
> ### Instruction:
> {instruction}
>
> ### Response:
> {response}

The losses are only computed for the tokens in responses. This template is also used for the instruction-response pairs in the information update training.

**Self Instruction Generation**   This prompt is given to the language model to be updated for self data creation. This prompt instructs the model to generate instructions from the information updating corpus.

> Given the input below, generate at least 5 questions that are directly related to the content of the input. Ensure that each question you generate does not contain coreferential words or pronouns (e.g., he, she, it, this, they, etc.). The questions should be clear, concise, and pertain specifically to details mentioned in the input.
> {Context}

The {Context} slot is filled with each individual news article from the information update corpus.

**Self Answer Generation**   This prompt is given to the language model to be updated for self data creation. This prompt instructs the model to generate responses for the instructions in the previous step from the information updatingcorpus.

> Answer the question based on the facts from the input. If there is no relevant information in the input, answer 'None'. Question: {Instruction} {Context}

---

[9]`https://github.com/tatsu-lab/stanford_alpaca`

The {Context} slot is filled with each individual news article from the information update corpus. The {Instruction} is from the outputs of last step. To ensure the generated instruction-response pairs pertain to the corpus, we remove those pairs when the response is None.

**Fact Finetuning Training Data**    This is the inputs to train the Fact Fine-tuning baseline in the main text. It is just the news articles.

> {News Article}

**Naïve Distillation**    This is the inputs to the train the Naïve Distillation Baseline. Only losses on the tokens after "Response" is used for training.

> Below is an instruction that describes a task. Write a response that appropriately completes the request.
>
> ### Instruction:
> {Instruction}
>
> ### Response:
> {Response}

Here the {Instruction} and {Response} are paired outputs from Self Instruction Generation and Self Answer Generation.

**Context-aware Distillation**    This is the inputs to the train the Naïve Distillation Baseline. Only losses on the tokens after "Response" is used for training.

> Below is an instruction that describes a task. Write a response that appropriately completes the request.
>
> ### Instruction:
> {Instruction}
>
> ### Response:
> The instruction is related to the following information: {News Article}. The response to {Instruction} is: {Response}

Here the {Instruction} and {Response} are paired outputs from Self Instruction Generation and Self Answer Generation. {News Article} is the corresponding news article from the information update corpus. Note that for unrelated instructions, the {News Article} is filled with "None". We repeat the instruction one more time to compensate for the limited sequence length and reduce the possibility of instructions being truncated. We think it may not be necessary to repeat the instruction if the computational resources supports sufficiently long training sequences. Only losses on the tokens after "Response" is used for training.

**Evaluation Data Generation**    We generate *CNN News* evaluation data using GPT-4. This prompt is given to GPT-4 to generate instruction-response pairs.

> Generate some questions[10] with answers related to facts from the following paragraph. Make sure each question is self-contained and specific enough for readers to associate it with the information provided in the paragraph, rather than confusing it with other similar events. Avoid using words such as "these", "this", or "the event", "the movie" referring to concepts not mentioned in the question. Please generate in the format of "1. Question: ... Answer: ..." {News Article}.

Because we strictly required the format of the generation in the last sentence, it is easy to parse the output pairs.

---

[10]In this work, we focus on instruction-response pairs in a question-answering format

