# OpenReview forum: "Exposure Bias Mitigation for Self Information Updating of Large Language Models"
_ICLR.cc/2024/Conference — ICLR 2024 Conference Withdrawn Submission_

### Official Review · Reviewer_fBKp · 2023-10-29

**Soundness:** 1 poor
**Presentation:** 1 poor
**Contribution:** 2 fair
**Rating:** 3
**Confidence:** 4

**Summary:**

The paper studied the task of updating an LLM with new knowledge. The proposed method turns a corpus containing new information into instruction (question) and response pairs ready for fine-tuning. Existing LLM is used to do this task. Authors argue that existing LLM has a tendency of using previously trained knowledge than the new one, aka exposure bias. The proposed solution is to repeat the relevant new content before answering the question in model-generated fine-tuning data.

Experiments are done with an instruction-tuned LLAMA-7B and two new corpora (CNN, NQ). Model-based evaluation showed that the proposed method outperforms the baseline model and simple fine-tuning (adding facts directly and adding q&a pairs without repeating the new facts). To see how the model would forget old knowledge after fine-tuning, experiments with different sizes of samples from existing instruction tuning sets are done, which shows that even a small set of old instructions can prevent significant forgetting.

**Strengths:**

How to update the model with new knowledge is an interesting problem.

**Weaknesses:**

Reviewer orders the concerns by their importance.

- Probably a wrong assumption about the limits of incorporating new information into an old model's response. Retrieval-augmented generation (RAG) is a widely-adopted practice now, as seen in ChatGPT's browser-plugin and Google Bard. It can equip the models with fresh information without the need of retraining. The assumption made in Section 4 paragraph #3, "However, it is impossible to maintain an infinitely large memory to store the new information" is not really a problem given modern search engines / information retrieval techniques.

- The hypothesis of exposure bias is not properly validated. Higher evaluation numbers of context-aware fine-tuning data over facts and facts+naive are not a sufficient proof that exposure bias exists. Reviewer suggests either design a more direct experiments at exposure bias, or do detailed analysis of the wins/losses between the results of different methods to show that the differences are indeed caused by exposure bias.

- Model-based evaluation is less convincing. All evaluations in the experiments are based on models. Reviewer thinks it's important to look at the results, either by human raters or with human analysis, to make sure the differences are not caused by the model's bias.

- Evaluation lacks confidence intervals. Some of the differences are quite narrow, e.g. in NQ. The CNN set only has ~300 evaluation prompts (smaller). The evaluator itself is a model. All the factors call for a confidence interval on all numbers to make sure they are statistically significant.

- Poor writing. Spelling errors and non-sense characters possibly from lack of proof-reading are common.

**Questions:**

Here are some detailed questions/suggestions.

First of all, there are numerous simple writing errors and misspellings. E.g.

- Last paragraph in Section 1, "Additionally, we perform an additional study", only one "additional" is needed.
- Section 1 last paragraph last sentence (before "To summarize"). It's too long to be understandable.
- Second claim at the end of Section 1: what is "mbbvpropose"?
- Definition 2.4. "Informtaion"?
- Right before Equation 5, what is "vcvc?"

Reviewer will not try to enumerate all of them. Some professional proofreading is needed (maybe using an LLM to help spell check, at least?)

Other non-writing related questions

- Please explain any new math notions when they first appear. E.g. section 2.2 first paragraph, X_T? Or in Definition 2.4 I_s(C)?

- Right after Definition 2.3, the "here P(r|A,i,T)", it doesn't match anything in Equation (3).

- Equation 7 is very confusing. What do you mean by indicator function and what do they do here?

- Table 1: adding the new prefix in the response will effectively change the model's output. Do you mean we will need to do post processing from then on?

- Page 5 footnote 1. Not clear what the authors are doing.

- Section 4 2nd paragraph. GPT-4 has 170 trillion parameters? That's a trillion with a T.  What is the source of information?

---

> ### Author Response · Authors · 2023-11-16
>
> Thanks for the review. Regarding the reviewer’s concern that the updating problem is not important since retrieval augmentation can be potentially done on infinitely large corpus, our arguments are
> Traditional tf-idf based retrievers (e.g., BM25) typically have inferior performance, while neural retrievers also need updating in essence. Moreover, even tf-idf based retrievers are very slow for very large corpus, and neural retrievers are even slower.
> As the example of Chat-GPT that the reviewer mentioned, it is worth noticing that Chat-GPT is also regularly updated with new training data. Its knowledge cutoff date was just updated to April 2023. We hope this fact can alleviate the reviewer’s concern on the lack of importance of this problem. We are happy to have further discussion over this concern,
>
> Weakness 2. We agree with the reviewer’s concern. We have provided a case study in the appendix that is related to the reviewer’s suggestion
>
> Weakness 3, we acknowledge that the model-based evaluation is not perfect, yet it outperforms traditional automatic metrics in terms of the alignment with human preferences. The reviewer also suggested human rating, however the cost of huma evaluation is an important factor under consideration. We are considering doing a small scale human evaluation on a subset of the evaluation data to demonstrate the consistency of the model scores and human ratings. We will keep you posted about the progress.
>
> Weakness 4 (lacking confidence interval), we actually put it in the appendix to save space.
> For the reviewers’ question 1,we have added the explanation of X_T which is the set of related text sequences.
> For the reviewer's confusion on the notation I_S(C) in definition 2.4, actually definition 2.4 is defining I_S(C). We have made this clearer.
> For reviewr’s question on the indicator function, for a logical statement p(x), an indicator function I(p(x)) takes the value of 1 if p(x) is true otherwise 0.
> Q4, we mentioned in section 3.5 that we did automatic post-processing.
> Q5, the footnote says that  the question is repeated in the response during training due to potential left truncation limited by the maximum sequence length.
> Q6, this is a typo. It should be 1.7T.
> Also, thanks to the reviewer for pointing out typos. We have fixed the typos.

---

> > ### Comment · Reviewer_fBKp · 2023-11-22
> > **Thanks for the answers.**
> >
> > Thanks for all the answers. I still think 1) large scale retrieval is not easy but a mostly solved engineering problem. 2) some human in the evaluation is still necessary.
> >
> > Again, I do think teaching model new knowledge more efficiently is a good direction. Looking forward to more progress!

---

> ### Author Response · Authors · 2023-11-22
>
> Thanks for the reply. However, still regarding the first weakness. We are not sure if the reviewer is convinced by our rebuttal that large-scale retrieval (regardless of whether it is solved) is not the right or best way to “teach” models new knowledge. I think this point is the review’s major concern over our paper from the review and also what we were trying to explain.

---

> > ### Comment · Reviewer_fBKp · 2023-11-22
> >
> > Yes the reviewer is not totally convinced that large scale RAG is not a good way to incorporate new knowledge.

---

> > > ### Author Response · Authors · 2023-11-22
> > >
> > > We would be interested in receiving the reviewer's thoughts on our arguments apart from the efficiency of the large scale retrieval. We mentioned that the SOTA retrievers are based on neural models, which in essence requires updating.  Granted, the updating of the retriever models can adopt a different strategy, but our point is that the problem of updating is perpendicular to the incorporation of retrieval techniques.

---

### Official Review · Reviewer_z86B · 2023-11-03

**Soundness:** 2 fair
**Presentation:** 3 good
**Contribution:** 2 fair
**Rating:** 6
**Confidence:** 3

**Summary:**

Authors propose a solution to the novel problem of self-information update for LLMs, without human intervention. This is a very relevant problem in the context of LLMs, as it’s important for LLMs to be up to date with the latest knowledge, and the current solutions involve training the model from scratch, or fine-tuning the existing model with the latest data, which can lead to catastrophic forgetting. Authors argue that fine-tuning can also lead to a situation where LLM ignore the new facts and knowledge, because of the exposure bias. A simple and practical technique is then proposed to handle the exposure bias problem.

**Strengths:**

- Authors propose a simple solution to the important problem of updating LLMs with the latest information. Keeping the LLMs up to date with the latest facts and knowledge is essential for its’ practical use-cases as a personal assistant and other applications. Existing commercial LLMs take months before they are updated with the latest information from the web.
- The solution proposed is simple to implement. It involves forcing the LLM to generate the news article first, followed by generating the response next.
- Experimental results from fine-tuning the LLaMA-7B model demonstrate the effectiveness of the method on different metrics and different datasets.

**Weaknesses:**

- The theoretical analysis is not very rigorous, but I don't see this as a major problem, and in fact authors have addressed this in the manuscript itself (Remark after Definition 2.4).

**Questions:**

- In the Figure 2 (a), the performance after fine-tuning drops somewhere between [2K, 3K] replay examples, but then improves post 4K examples. How would someone applying this technique choose the replay buffer size to avoid the drop in the performance below the base model?

---

> ### Author Response · Authors · 2023-11-16
> **Response on the Strategy of Picking Replay Examples**
>
> Thanks for recognizing the importance of the problem and the effectiveness of our method. The reviewer’s question on Figure 2(a) is a very interesting question that we are also investigating after the submission of this work. Although this investigation is still in progress, our current understanding is that in addition to the size,  the selection and the training order of the replay samples seem to be important for quality of learning. Our observation is that when the samples are not arranged properly, the model tends to learn negative associations (connect evaluation questions to unrelated information). However, it is worth mentioning that we focus more on the challenge of exposure bias during updating for this work .We believe our ongoing work will provide a better understanding of the question the reviewer poses.

---

> > ### Comment · Reviewer_z86B · 2023-11-18
> > **Thanks for the rebuttal**
> >
> > Thanks for your rebuttal. I acknowledge that I have read it.

---

### Official Review · Reviewer_f8AU · 2023-11-09

**Soundness:** 3 good
**Presentation:** 3 good
**Contribution:** 3 good
**Rating:** 6
**Confidence:** 3

**Summary:**

This work addresses the problem of continual fine-tuning of LLMs using new training corpora. In contrast to previous approaches that involve significant human effort in converting unstructured data into more structured data to make the best use of new training corpora in fine-tuning LLMs, this work attempts to eliminate the human involvement in the process. It proposes a self information updating task for LLMs that makes use of unstructured new training corpora in the fine-tuning of LLMs without requiring human editorial intervention. The self information updating task is formulated as a self-distillation task with the LLM as the teacher and the new training corpora as the context. Specifically, this consists of first generating instruction response pairs from the new training corpora by using the LLM and then using the resulting pairs for fine-tuning. However, this naive approach is plagued by the issue of exposure bias where existing information from LLM is prioritized over the novel information in the new training corpora. This issue is identified and analyzed theoretically and it is noted that the bias affects both response generation and probability of instructions.

The work proposes a heuristic for mitigating exposure bias. The key idea is to do a context-aware distillation which is essentially identifying the source of the instruction (new corpus or pretraining corpus) and making use of only probability terms that are relevant for the source. Given a training triple (i, r, d), where d is the article from the new corpus from which the LLM generated the instruction-response pair (i, r), the model is forced, during finetuning, to learn to generate d from i and then r from (d, i). Here d serves as the context for i to the LLM for generating i.

The work presents results from experiments on two datasets - CNN News and NQ Val. The first data set consists of a small news corpus (50 CNN news articles from the period March and April 2023) with a resulting evaluation set of 301 instruction-response pairs generated using GPT-4. The NQ Val data set consists of extracted paragraphs from Wikipedia pages (long answers for questions in the validation split of the Natural Questions benchmark) with an evaluation set consisting of questions and short answers. As the baseline an instruction-following model from the LLaMA-7B fine tuned with instruction-following data from Alpaca and InstructionWild is used. (Of these, a randomly sampled 300 instruction-response pairs is used as old data.) The work answer consistency and context consistency as the metrics for evaluating the proposed approach as well as the baseline. Experimental study reveals that the proposed approach produces significant improvement over the baseline on both metrics for instruction-response pairs from new corpus while giving marginally negative improvement for instruction-response pairs from the instruction fine-tuning corpus. In contrast, a combination of principled but subject to exposure bias methods (namely, Fact + Naive) gives slightly lower improvement for instruction-response pairs from new corpus while not degrading performance on instruction-response pairs from the instruction fine-tuning corpus.

**Strengths:**

1. Addresses an interesting and relevant problem in the context of finetuning LLMs.
2. Provides interesting and useful theoretical analysis that highlights the source of exposure bias problem.

**Weaknesses:**

1. Dropping completely probability terms based on context (to get Equation 7 from Equation 5) no doubt prioritizes novel information in the new training corpus over existing information from LLM for instruction-response pairs generated from the new corpus. But this also affects the model's performance on old instruction-response pairs as evident from Table 2. A possibly better approach is to use all the terms in Equation 5 but with source dependent weight that is determined by hyperparameter tuning.

2. While the idea of generating instruction-response pairs from LLM for new corpus is interesting and eliminates the need for human effort, it is also a potentially problematic issue as current LLMs are known to hallucinate and as a consequence the instruction-response pairs could be affected by it.

**Questions:**

1. In Table 2, the answer consistency score of Context-aware is significantly higher than that of the baseline for new corpus. This is very encouraging indeed. However, the performance of  Context-aware on new corpus is significantly lower than that of the baseline for Old (0.480 vs 0.699 for CNN News and 0.256 vs 0.699 for NQ Val). What explains this gap and what can be possibly done to reduce it?

2. In Table 2, the answer consistency score of the baseline for Old is nearly the same for both data sets (CNN News:0.696 and NQ Val:0.691). In contrast, Context-aware seems to be doing significantly better on CNN News than on NQ Val (0.480 vs 0.256). What explains this gap and what can be possibly done to reduce it?

3. How many instruction-response pairs are in NQ Val and how many are used in evaluation?

4. From Figure 2.a it appears that some forgetting is inevitable in the proposed approach and increasing the no of replay examples doesn't really help alleviate this problem beyond a point. One wonders if forgetting will become more severe when fine tuning is done continually and with more datasets than the two used in the experimental studies. Figure 2.b seems to indicating this (score for old reduces further when the model already fine-tuned with NQ Val is further fine tuned with CNN News). What can be possibly done to address this issue?

---

> ### Author Response · Authors · 2023-11-16
> **Response to the Reviewer's Questions and Discussion on Open Problems the Reviewer Posed**
>
> Thanks for recognizing the importance of the problem and the effectiveness of our method. Regarding the reviewer’s question on Table 2 (Q1 and Q2)., we have two observations. Firstly, on the CNN news corpus where the evaluation QA pairs are generated based on the information updating corpus, the context consistency score is close to the score on old instructions. Note that the reason we report context consistency is that sometimes the GPT-4 generated answers are brief and more detailed answers generated by our model are penalized for additional details.Therefore we think in the case of CNN News the performance gap is actually small. For NQ Val, we believe the reason lies in the different approach to create the dataset. In NQ, questions are first created without referring to the corpus. Then annotators try to associate relevant wikipedia articles to the questions. This might lead to some challenging cases where more complex reasoning is required to use the information in the document to answer the question. Moreover, we filter our questions where the base model without fine-tuning achieves higher scores in NQ Val, which makes the remaining questions more probable to be challenging questions. In general, we believe this is a good question and deserve deeper investigation.
>
> Regarding the reviewer’s Q3, there are 8k examples in the original validation set. We remove those with empty answers and those the base model achieves high scores without fine-tuning, resulting in 1382 examples. Many examples are filtered because LLaMA is very likely to have been pre-trained on Wikipedia pages.
>
> For Q4, we have a similar understanding of the forgetting problem as the reviewer. For the scope of this paper, we focus more on the quality of updates and the experiments on the continual setting serves as an initial exploration in the forgetting problem. Also the continual experiments shows that our method’s requirements on the replay tokens is potentially smaller than traditional fine-tuning methods.

---

> > ### Comment · Reviewer_f8AU · 2023-11-20
> > **Author response**
> >
> > I thank the authors for their response and acknowledge that I've read it.

---

### Official Review · Reviewer_b44p · 2023-11-10

**Soundness:** 3 good
**Presentation:** 2 fair
**Contribution:** 3 good
**Rating:** 5
**Confidence:** 4

**Summary:**

The manuscript presents a comprehensive study of a novel approach to updating large language models (LLMs) with the latest information without requiring substantial human intervention. Recognizing the limitations of LLMs in accessing up-to-date information due to reliance on pretraining corpora, the authors propose a self-information updating task. The study also examines the issue of knowledge forgetting, proposing a solution with a compact replay buffer.

**Strengths:**

1、The study introduces a novel and practical method for updating LLMs with the latest information, which is a significant advancement in the field.

2、The research delves into the challenge of knowledge forgetting, offering a promising solution with a replay buffer mechanism.

3、The experimental results showing improvements in factual consistency scores are impressive and indicative of the method's efficacy.

**Weaknesses:**

1、The manuscript could provide a more in-depth analysis of how the integration of facts into training losses specifically counteracts exposure bias.

2、This paper only conducted experiments on factual consistency, which is not convincing enough. Furthermore, an explanation has not been provided as to why this task was chosen to demonstrate the effectiveness of the method for updating large language models (LLMs) with the latest information.

3、There are many unclear aspects in the formula explanations within the paper, leading to confusion. For example, it is not specified what the text sequence \( x \in X \) in the context before and after Equation 1 specifically represents.

Overall, the motivation of this paper and the explanations of the symbols used throughout the text are not very clear, and the expression still needs to be more clearly and concisely articulated.

**Questions:**

How does the setting of updating large models with new corpora proposed in this paper differ from continual learning and model editing? The paper lacks discussion and analysis in this regard.

---

> ### Author Response · Authors · 2023-11-16
> **Response to the Concern over Evaluation and Some Confusions**
>
> Thanks for recognizing the importance of the problem and the effectiveness of our method. Regarding the reviewer’s question on the different with model editing, existing methods typically use human-curated corpus for editing, where the knowledge to edit is carefully cleaned into either triples or simple sentences containing the fact  (e.g., [2307.12976] Evaluating the Ripple Effects of Knowledge Editing in Language Models (arxiv.org)). In this work, we try to avoid human involvement in the process and perform updating on raw text. For the connection with continual learning, the setting of this work is a subset of continual learning focusing on the information in the corpus. This is also the reason why we evaluate with QA pairs querying the information in the updating corpus instead of using perplexities for evaluation. We have a more brief discussion in the current related work section. The above explanation is also related to the reviewer’s concern over the evaluation using factual consistency and the QA task in weakness 2.
>
> Regarding the reviewer’s confusion on the meaning of $x$ after equation 1, we shift to an input-output (i,r) format for language modeling because our experiments are conducted for instruction-following models. We have added more explanations to avoid the confusion. The reason we use instruction-following models is due to the difficulty of evaluating information acquisition using QA pairs for base models without instruction-following abilities. Moreover,  We would like to point out that in auto-regressive language models, language modeling can also be rewritten in an input-output format P(x_(1:n+m)) = P(x_(1:n)P(x_(n+1:n+m)|x_(1:n)) by picking an arbitrary breakpoint.
>
> Please let us know if you have any further concerns and confusions.

---

### Official Review · Reviewer_o5o3 · 2023-11-11

**Soundness:** 3 good
**Presentation:** 2 fair
**Contribution:** 3 good
**Rating:** 6
**Confidence:** 3

**Summary:**

This paper introduces a task called Self Information Updating (SIU) in LLMs, which requires the model to update itself using unstructured information sources, without the need for human intervention, resulting in a more practical task. Additionally, the paper introduces an approach, context-aware distillation, to address exposure bias, which tends to prioritize old information over new data when updating the LLMs. Finally, the effectiveness of this approach is demonstrated through evaluation using two new created datasets, revealing a notable 0.16 increase in the factual consistency score.

**Strengths:**

1- Reproducibility: The authors mentioned that their source code and data will be shared once their work is published. This means anyone can use these to reproduce their results.
2- Two new datasets were developed to evaluate information updates.
3- Factual consistency is increased by up to 0.16 using their new proposed approach.
4- The forgetting problem is addressed to some extent by using a small portion of the original training data.

**Weaknesses:**

How big can be the information updating corpus? If we have big information updating
corpus, how it will affect the performance?

**Questions:**

Please see the weakness section.

---

> ### Author Response · Authors · 2023-11-16
> **Discussion on the size of updating corpus**
>
> Thanks for recognizing the importance of the problem and the effectiveness of our method. Regarding the reviewer’s concern over the size of the updating corpus, for the scope of this paper the maximum size we used for updating is ~1.6k documents. Although we haven’t experiment with our method on larger corpus limited by the computational capacity, we would like to point out that the problem that our proposed method is solving, i.e. failures of LLMs from directly fine-tuning on the facts or instruction-response pairs will not vanish by increasing the size of the corpus. This problem is supported by several papers released after the submission of this work on arxiv (e.g., [2307.12976] Evaluating the Ripple Effects of Knowledge Editing in Language Models (arxiv.org), https://arxiv.org/pdf/2309.14316.pdf), as well as some failures we observed from GPT-4.